# The *AtMINPP* Gene, Encoding a Multiple Inositol Polyphosphate Phosphatase, Coordinates a Novel Crosstalk between Phytic Acid Metabolism and Ethylene Signal Transduction in Leaf Senescence

**DOI:** 10.3390/ijms25168969

**Published:** 2024-08-17

**Authors:** Xiaoyun Peng, Haiou Li, Wenzhong Xu, Qian Yang, Dongming Li, Tingting Fan, Bin Li, Junhui Ding, Wenzhen Ku, Danyi Deng, Feiying Zhu, Langtao Xiao, Ruozhong Wang

**Affiliations:** 1Hunan Provincial Key Laboratory of Phytohormones and Growth Development, Hunan Agricultural University, Changsha 410128, China; pengxiaoyun@hncu.edu.cn (X.P.); yangqian120324@163.com (Q.Y.); ste0622@163.com (T.F.); djhyi@126.com (J.D.); kwz9802@126.com (W.K.); feiyingzhu@hunaas.cn (F.Z.); 2State Key Laboratory of Plant Diversity and Specialty Crops, Institute of Botany, Chinese Academy of Sciences, Beijing 100093, China; xuwzh@ibcas.ac.cn; 3Key Laboratory of Herbage & Endemic Crop Biology of Ministry of Education, School of Life Sciences, Inner Mongolia University, Hohhot 010021, China; lidongming0118@163.com; 4Hunan Academy of Agricultural Sciences, Changsha 410125, China; binli369@hnu.edu.cn

**Keywords:** Arabidopsis, EIN3, ethylene, leaf senescence, multiple inositol polyphosphate phosphatase, senescence-associated genes

## Abstract

Plant senescence is a highly coordinated process that is intricately regulated by numerous endogenous and environmental signals. The involvement of phytic acid in various cell signaling and plant processes has been recognized, but the specific roles of phytic acid metabolism in Arabidopsis leaf senescence remain unclear. Here, we demonstrate that in *Arabidopsis thaliana* the multiple inositol phosphate phosphatase (AtMINPP) gene, encoding an enzyme with phytase activity, plays a crucial role in regulating leaf senescence by coordinating the ethylene signal transduction pathway. Through overexpressing *AtMINPP* (*AtMINPP–OE*), we observed early leaf senescence and reduced chlorophyll contents. Conversely, a loss-of-function heterozygous mutant (*atminpp/+*) exhibited the opposite phenotype. Correspondingly, the expression of senescence-associated genes (SAGs) was significantly upregulated in *AtMINPP–OE* but markedly decreased in *atminpp/+*. Yeast one-hybrid and chromatin immunoprecipitation assays indicated that the EIN3 transcription factor directly binds to the promoter of *AtMINPP.* Genetic analysis further revealed that *AtMINPP–OE* could accelerate the senescence of *ein3–1eil1–3* mutants. These findings elucidate the mechanism by which AtMINPP regulates ethylene-induced leaf senescence in Arabidopsis, providing insights into the genetic manipulation of leaf senescence and plant growth.

## 1. Introduction

Leaf senescence is a programmed process that involves a series of cytological and biochemical changes. During senescence, there is an orderly transformation in the structure, metabolism, and gene expression of leaf cells [1,2,3]. In the early stages of senescence, leaves exhibit degreening characteristics such as chlorophyll degradation, reduced photosynthetic activities, as well as decreased levels of total RNA and proteins [4]. Additionally, the expression of numerous SAGs and transcription factors is significantly altered [5,6,7,8,9]. Leaf senescence represents the final stage of leaf development and is a highly regulated process influenced by plant hormones and environmental factors [10,11]. Phytohormones play crucial roles in regulating leaf senescence. Ethylene, abscisic acid, and jasmonic acid are known to promote senescence, while cytokinins delay it [12]. Among these hormones, ethylene has been widely recognized as a key regulator of leaf senescence, functioning as a gaseous plant hormone [11,13]. Exogenous ethylene treatment accelerates leaf senescence, whereas the application of inhibitors that block ethylene biosynthesis delays it [14,15]. The biosynthetic and transduction pathways of ethylene have been extensively described. Two enzymatic steps are involved in ethylene synthesis: S-AdoMet is converted to 1-aminocyclopropane-1-carboxylate (ACC) by ACC synthase, and ACC oxidase (ACO) further oxidates ACC to form ethylene [16]. EIN3, a crucial transcription factor, acts as a positive regulator for the ethylene-responsive genes. The functional loss of EIN3 and its close homolog EIL1 leads to delayed leaf senescence under natural conditions, darkness, and ethylene-induced conditions [17,18]. 

Phytic acid (PA), also known as myo-inositol-1,2,3,4,5,6-hexakis-phosphate (InsP_6_), is the predominant storage form of phosphorus (P) found in plants [19,20,21]. In addition to its role in phosphorus storage, phytic acid is involved in various metabolic functions, such as the storage of other mineral elements, a co-factor of the auxin receptor, and RNA transportation or DNA metabolism [22,23]. However, a particular interest is the fact that phytic acid typically constitutes 60–80% of the total P in seeds, affecting the nutritional quality of food for both humans and livestock. Due to limited phytase activities in the digestive tract, non-ruminant animals cannot utilize phytic acid effectively, leading to the excretion of almost 90% of phytate from animal feed into water bodies, eventually causing water pollution and mineral deficiencies [24,25,26,27]. The significant implications for human well-being have spurred a vast body of literature on phytic acid in seeds and fruits [28,29,30], with substantial efforts directed towards breeding low phytic acid crops [24,25,31,32] or crops capable of producing heterologous phytases [33].

Phytases, also known as myo-inositol hexakisphosphate phosphohydrolases (EC 3.1.3.8, EC 3.1.3.26, and EC 3.1.3.72), are phosphatases that can initiate the degradation of phytate by the removal of one or more phosphate groups. The less specific phosphatases then proceed to hydrolyze the remaining phosphate groups ever since [5,34]. According to their catalytic mechanisms, phytases can be categorized into histidine acid phosphatases (HAPs), purple acid phosphatases (PAPs), cysteine phosphatases (CPs), and β-propeller phytases (BPPhys). Most plant phytases are HAPs, a very large group of acid phosphatases: the N-terminal active site motif RHGXRXP and the C-terminal motif HD form the catalytic site of these phosphatases [35,36,37]. Multiple inositol polyphosphate phosphatases (MINPPs), a typical phosphatase group from HAPs, appear to be localized in the endoplasmic reticulum (ER) and catalyze the hydrolysis of InsP_6_, InsP_5_, and InsP_4_ in mammals, which are assumed to have a central role, not only in providing bioavailable phosphate to growing cells but also in the production of important physiologically downstream metabolites of inositol phosphates [38]. Interestingly, it has been demonstrated that phytases belonging to the histidine phosphatase family possess multiple inositol polyphosphate phosphatase (MINPP) functionalities in several plant species [37,39]. However, the phytase gene in Arabidopsis has not yet been identified, let alone its role in leaf senescence.

In this study, we reported that a *multiple inositol phosphate phosphatase* (*MINPP*) gene, which encodes a phytase in *Arabidopsis thaliana*, plays an important role in leaf senescence. Additionally, we discovered that the EIN3 transcription factor can directly bind to the promoter of *AtMINPP*, thereby activating its transcriptional expression. This coordination establishes a crosstalk between ethylene signal transduction and phytic acid metabolism, ultimately accelerating leaf senescence in Arabidopsis.

## 2. Results

### 2.1. Isolation and Identification of the Multiple Inositol Polyphosphate Phosphatase Gene AtMINPP in Arabidopsis

We initially used BLAST to identify At1G09870 as a potential homolog of multiple inositol-polyphosphate phosphatases (MINPPs) in the *Arabidopsis thaliana* genome and named it *AtMINPP*. The search was conducted on UniProt (https://www.uniprot.org, accessed on 26 February 2024) and HomoloGene (https://www.ncbi.nlm.nih.gov/homologene/37980, accessed on 26 February 2024). A phylogenetic tree generated by the bootstrap method revealed that AtMINPP was clustered on a branch together with wheat, barley, lily, rice, and maize MINPP, HAP proteins, and AP proteins (Figure 1A). The AtMINPP protein was aligned with known histidine acid phosphatases (HAP), acid phosphatases (AP), and MINPPs from other plants. Most of the clones contain an RHGTRAP motif, which was in accordance with the HAP active site motif, RHGXRXP. However, AtMINPP exhibited an S instead of an A in the conserved motif. The HAE sequence, as the second identifier of the HAP motif, was found in all plants. While most of the clones showed the conservative C-terminal motif HD, AtMINPP and TaPhylla had an A and E instead of a D, respectively (Figure 1B). The full-length coding sequence (CDS) of *AtMINPP* was cloned into the pET28a–MBP vector to realize the overexpression of the protein in vitro. An amount of soluble recombinant protein was obtained, and the activity of acid phosphatase and phytase in AtMINPP was 1.68 U/L and 516.33 U/g, respectively (Figure 1C,D).

The expression patterns of *AtMINPP* were explored during seedling growth by detecting the *AtMINPP* promoter activity. According to GUS staining, the *AtMINPP* promoter was highly active in both roots and cotyledons, as well as in other tissues, such as young stem tips, whole young true leaves, flower buds, pedicel nodes, and young siliques. These results indicated that *AtMINPP* was widely expressed in vegetative and reproductive organs of *Arabidopsis thaliana* (Figure 2A–F), with the highest expression in the leaves in the 30-day-old Col–0 seedlings (Figure 2G).

To determine the subcellular localization of AtMINPP, the CDS was fused to the C-terminus of green fluorescence protein (GFP). The constructed plasmid, along with the endoplasmic reticulum localization marker DEP2-RFP plasmid, were co-transferred into wild-type Arabidopsis protoplasmic cells. Under a laser confocal microscope, the fusion of two signals was observed, indicating that AtMINPP was located in the endoplasmic reticulum (Figure 2H).

To further investigate the function of AtMINPP, we characterized the T-DNA insertion mutant (CS400603) from the Arabidopsis Information Resource (TAIR, https://www.arabidopsis.org, accessed on 26 February 2024). It was shown that the phytic acid content was highly elevated due to a defect of *AtMINPP* (Appendix A). Thus, we suggested that *AtMINPP* (At1G09870) encodes a histidine phosphatase similar to multiple inositol polyphosphate phosphatases.

### 2.2. AtMINPP Involves the Regulation of Leaf Senescence

To elucidate the biological function of AtMINPP, the overexpression lines (referred to as *AtMINPP–OE*) were constructed by expressing *AtMINPP* under the *CaMV 35S* promoter. Additionally, using the CRISPR/Cas9 technique, we generated heterozygous mutant lines with different base deletions (2 bp and 43 bp) at the target site, which exhibited curly and dwarf rosette leaves after 4 weeks (Appendix A). Interestingly, the F_3_ hybrid progeny derived by crossing the mutant carrying a 43 bp deletion with Col–0 exhibited homozygous lethality and the same phenotype.

In our study, the phenotype analysis revealed several important findings. *AtMINPP–OE* (Appendix A) exhibited earlier leaf senescence compared to Col–0, while the loss-of-function heterozygous mutant of *AtMINPP*, known as *atminpp/+* (Appendix A), displayed delayed leaf senescence compared with Col–0 (Figure 3A). When comparing the rosette leaves of 6-week-old plants, *AtMINPP–OE* showed more yellow leaves, whereas *atminpp/+* exhibited fewer yellow leaves compared to Col–0 (Figure 3B). Furthermore, the pigment examination showed that *AtMINPP–OE* exhibited a faster reduction in the chlorophyll content after 5 weeks, whereas *atminpp/+* consistently maintained more chlorophyll compared to Col–0 (Figure 3C). The ion leakage rate of *AtMINPP–OE* noticeably increased compared to Col–0, whereas that of *atminpp/+* exhibited a notable decrease (Figure 3D). Additionally, the photochemical efficiency of photosystem II (PSII) using the Fv/Fm ratio revealed a significant decrease in *AtMINPP–OE* compared to Col–0, while *atminpp/+* displayed a remarkably higher Fv/Fm ratio (Figure 3E). Meanwhile, compared to Col–0, the phytic content in *AtMINPP–OE* decreased slightly, while *atminpp/+* displayed a higher phytic content (Figure 3F). Simultaneously, the phytic content of leaves in the early senescent leaves was lower than that of the nonsenescent leaves but higher than that in the late senescent leaves (Figure 3G). These findings suggested that phytic acid metabolism regulated by AtMINPP is involved in leaf senescence.

To validate the involvement of AtMINPP in leaf senescence, we examined the gene expression profile in different genetic backgrounds of AtMINPP. Firstly, the transcript levels of SAGs in 5-week-old leaves of Col–0, *AtMINPP–OE*, and *atminpp/+* were detected by RT–qPCR. As shown in Figure 4A, the expression levels of *SAG12*, *SAG13*, *SAG29*, *SAG113*, *SAG201*, *ANAC047*, *SENESCENCE1* (*SEN1*), *SEN4, SENESCENCE–INDUCED RECEPTOR–LIKE KINASE* (*SIRK*), and *BFN1* were significantly higher in *AtMINPP–OE* compared to Col–0, while the expression levels of these genes were noticeably lower in *atminpp/+*. Furthermore, RT–qPCR analysis revealed that the expression of *AtMINPP* gradually increased from the base to the tip of a senescing leaf, which resembled the expression pattern of the senescence marker gene *SAG13* (Figure 4B). In leaves at different developmental stages, the transcript levels of *AtMINPP* were the lowest in nonsenescent leaves, which increased in early senescent leaves, and reached the highest levels in late senescent leaves, mirroring the trend of the senescence marker gene *SAG13* (Figure 4C). These findings provided compelling evidence for the pivotal role of AtMINPP in the regulation of SAGs. 

### 2.3. AtMINPP Mediates the Ethylene Signaling in Leaf Senescence

To examine the effect of ethylene on the expression of *AtMINPP*, we employed two inhibitors of ethylene synthesis, namely AgNO_3_ and aminoethoxyvinylglycine (AVG) [40]. The rosette leaves from 4-week-old seedlings of Col–0, *AtMINPP–OE*, and *atminpp/+* were detached and treated with AgNO_3_ and AVG. After 4 days of treatment, compared to the control, the yellowing process in both Col–0 and *AtMINPP–OE* leaves was significantly suppressed by AVG and AgNO_3_ treatments (Figure 5A). Notably, since *atminpp/+* leaves showed delayed senescence under control conditions, AgNO_3_ and AVG exhibited no significant effect on these leaves (Figure 5A). Furthermore, after the treatment of AgNO_3_ and AVG, the chlorophyll content and Fv/Fm ratio of *AtMINPP–OE* were noticeably higher than those of the control, while the ion leakage rate was lower than that in the control, in accordance with the senescent phenotypes under the AgNO_3_ and AVG treatment (Figure 5B–E). These results suggested that inhibiting ethylene synthesis can delay the AtMINPP-mediated acceleration of leaf senescence.

In order to confirm the involvement of AtMINPP in ethylene–mediated leaf senescence, we examined the response of *AtMINPP* to ethylene in the 2-week-old Col–0 seedlings grown on plates. As shown in Figure 5F, the expression of *AtMINPP* was significantly induced by ACC treatment, which began to increase at 12 h and peaked after 24 h treatment, indicating its involvement in ethylene-regulated leaf senescence. Meanwhile, the fifth to eighth rosette leaves from 4-week-old Col–0, *AtMINPP–OE*, and *atminpp/+* plants were detached and treated with ACC. After 60 h treatment, the leaves of *AtMINPP–OE* exhibited more pronounced yellowing than those of Col–0, while *atminpp/+* displayed a stay–green phenotype in comparison to Col–0 and *AtMINPP–OE* (Figure 5G). Measurement of the chlorophyll content indicated a faster decline in chlorophyll levels in *AtMINPP–OE* leaves under ACC treatment, while *atminpp/+* slightly decreased compared with the control plants under ACC treatment (Figure 5H). Nevertheless, the relative chlorophyll degradation ratio of *atminpp/+* was much lower than Col–0 (Figure 5I). This finding suggested that a defect of *AtMINPP* can lead to decreased sensitivity of the *atminpp/+* to ACC. The ion leakage rate of *AtMINPP–OE* was much more elevated than those of Col–0 and *atminpp/+* under ACC treatment (Figure 5J). Additionally, the Fv/Fm ratio revealed a more significant decrease in *AtMINPP–OE* leaves compared to Col–0 and *atminpp/+* after ACC treatment (Figure 5K). These findings suggested that AtMINPP plays a vital role in ethylene–induced leaf senescence.

### 2.4. EIN3 Binds to the Promoter of AtMINPP Both In Vitro and In Vivo

To identify the upstream regulator for *AtMINPP*, a yeast one–hybrid screening assay was conducted using the promoter of *AtMINPP* as the bait. This screen identified several transcription factors that could physically bind to the *AtMINPP* promoter, including EIN3. The growth experiment revealed that only yeast clones containing the *AtMINPP* promoter and *EIN3* gene could survive on the selection medium (SD/–leu/AbA100) (Figure 6A,B), suggesting that EIN3 can directly bind to the promoter of *AtMINPP* in vitro. Subsequently, an in vivo chromatin immunoprecipitation (ChIP) assay was performed to investigate the interaction between EIN3 and the *AtMINPP* promoter by using the *35S::EIN3–MYC* transgenic line. Upon scanning the *AtMINPP* promoter, three putative EIN3 binding sites (EBS) were discovered, namely TACAT and TTCAA, located at 303~307 bp, 481~485 bp, and 677~681 bp upstream of the gene’s start codon (ATG) (Figure 6C). The ChIP assay indicated that the abundance of the three regions from the *35S::EIN3–MYC* transgenic line was notably more enriched than that from Col–0, revealing that EIN3 can bind to the *AtMINPP* promoter region (Figure 6D). Then, the expression of *AtMINPP* in Col–0 and *ein3–1eil1–3* mutants was examined. As shown in Figure 6E, the expression level of *AtMINPP* was significantly suppressed in *ein3–1eil1–3* compared to Col–0, which was similar to the expression patterns of the senescence marker gene *SAG13*. These findings suggested that EIN3 positively regulates *AtMINPP* expression by directly binding to the EBS sites on its promoter during ethylene–mediated leaf senescence in Arabidopsis.

### 2.5. AtMINPP Mediates Ethylene Signaling to Accelerate Leaf Senescence

To verify the genetic relationship between EIN3 and AtMINPP in regulating leaf senescence, *AtMINPP–OE* and *ein3–1eil1–3* mutants were crossed, and a double mutant *ein3–1eil1–3* overexpressing *AtMINPP* was obtained from F_3_ progeny. Notably, *AtMINPP–OE* prominently promoted leaf senescence in the *ein3–1eil1–3* mutant background (Figure 7A). The chlorophyll content (Figure 7B) and Fv/Fm ratio (Figure 7D) of *ein3–1eil1*–3 were significantly higher than those of *ein3–1eil1–3*/*AtMINPP–OE* and *AtMINPP–OE.* Moreover, the ion leakage rate of the double mutant was higher than that of *ein3–1eil1–3* but lower than that of *AtMINPP–OE* (Figure 7C). These results strongly indicated that EIN3 enhances ethylene signaling in leaf senescence by positively regulating the expression of *AtMINPP.*

In brief, our results demonstrated that *AtMINPP*, functioning as a phytase gene directly regulated by EIN3, plays a crucial role as a bridge linking phytic acid metabolism and ethylene signal transduction in leaf senescence.

## 3. Discussion

A significant amount of scientific research has primarily focused on the presence of phytic acid in seeds and fruits [41]. In 2009, Winkler and Zotz were the first to investigate the phytic acid content in wild epiphytic pineapples, particularly in plants with long-lived leaves from nutrient-deficient habitats. They further proposed a potentially essential ecological role of phytic acid in leaves [23,42]. Furthermore, it has been confirmed that relatively high concentrations of phytic acid have been found in the leaves of *Rivina humilis* L. [43]. Remarkably, our study observed variations in phytic acid content during leaf senescence (Figure 3G), revealing the biological significance of phytic acid metabolism in the final process of leaf development.

While phytases in seeds, leaves, and roots share significant similarities, the physiological function of phytases in non–reproductive tissues remains unclear [44,45]. Phytate may serve as a universal transitory form of phosphorus storage and inositol production in both reproductive and non-reproductive plant tissues. Regardless of their location, phytases may play a role in releasing inositol derivatives and free phosphate [46], as confirmed in our research. Our study unveiled a new HAP gene, *AtMINPP*, which exhibits both acid phosphatase activity and phytase activity and is highly expressed in *Arabidopsis thaliana* leaves (Figure 1C,D and Figure 2G). This finding aligns with previous reports indicating that AtPAP15 and AtPAP23, purple acid phosphatases, exhibit phytase activities in seedlings, germinating pollen, and flowers, respectively [47,48]. Interestingly, we discovered that *AtMINPP* has no other homologous gene in Arabidopsis, and the homozygous knockout mutant is lethal. This indicates that *AtMINPP* is essential for the growth and development of plants. Interestingly, the heterozygous *atminpp*/+ exhibited a mutant phenotype in our study, which is also present for other genes. Take *Cullin1*, for example, the rosette leaves of heterogeneous mutant *axr6–1*/+ and *axr6–2*/+ are shorter and more wrinkled than Col–0 [49]. Moreover, the heterozygous mutants we generated are valuable in understanding the mechanism of phytic acid metabolism and the biological significance therein of plants. Our analysis revealed that the phytic acid content in the leaves of *atminpp/+* plants increased by approximately 50% compared to the wild type (Figure 3F), indicating that AtMINPP is a distinctive phosphohydrolase capable of initiating the dephosphorylation of phytate.

Leaves typically undergo senescence in the late stages of their functional periods as a response to developmental signals and environmental stimuli [2]. With the advancements in molecular methods, significant progress has been made in understanding leaf senescence, a complex and highly regulated physiological process that is influenced by various internal and external factors such as plant hormones and environmental stress [50]. Traditionally, ethylene was believed to promote leaf senescence through a signaling cascade involving ETHYLENE INSENSITIVE 2 (EIN2), EIN3, and ORESARA1 (ORE1) in Arabidopsis [18,51,52,53]. EIN3, a transcription factor (TF) and critical senescence inducer, directly activates the expression of master senescence-associated genes *ORE1/NAC2* or binds to the promoters of *microRNA164* (*miR164*), repressing its transcription and upregulating the transcript levels of *ORE1/NAC2*, which is a target gene of *miR164*, in regulating leaf senescence [50]. In this study, we found that *AtMINPP* was highly expressed in the senescent part of a deteriorating leaf (Figure 4B), and more significantly, *AtMINPP* expression increased with leaf age, indicating a correlation between AtMINPP and leaf senescence. These results suggest that AtMINPP may play a crucial role in leaf senescence (Figure 4C). Therefore, overexpression of *AtMINPP* accelerated leaf senescence, whereas the loss of *AtMINPP* function resulted in delayed leaf senescence compared to Col–0 (Figure 3A,B). This observation aligns with findings that mutants like *pap26*, a purple acid phosphatase, exhibited a phenotype of delayed leaf senescence in Arabidopsis and rice [54,55]. Moreover, we screened the 4-week-old detached leaf cDNA library of *Arabidopsis thaliana* using the upstream promoter sequence of the *AtMINPP* gene as bait through yeast single hybridization, and EIN3 was subsequently identified as the upstream transcription factor of *AtMINPP* (Figure 6B). EIN3 then regulates the expression of *AtMINPP* both in vitro and in vivo (Figure 6 and Figure 7).

The precise regulation of plant senescence involves the coordinated expression of numerous SAGs. A significant advancement in our molecular understanding of this process has been made through the identification and characterization of hundreds of SAGs and various senescence-related mutants [5,56]. The expression levels of senescence-related genes, including *SAG12, SAG13, SAG29, SAG113, SAG201, BFN1, SEN1, ANAC047*, *SIRK*, and *SEN4*, were elevated in *AtMINPP–OE* but reduced in *atminpp/+* (Figure 4A). These findings further support the hypothesis that *AtMINPP* plays a role in regulating leaf senescence. In our research, we examined the phytic content in different genetic backgrounds of AtMINPP. It showed that the phytic content in *AtMINPP–OE* decreased slightly, compared to Col–0, while *atminpp/+* displayed a higher phytic content. We also detected the phytic content in different stages of development in the leaves. It showed that the phytic content of early senescent leaves was lower than that in the nonsenescent leaves but higher than that in the late senescent leaves (Figure 3F,G). Therefore, we speculate that the degradation of phytic acid by AtMINPP regulates the homeostasis of Pi in Arabidopsis, thereby mediating the activity of phosphatases and kinases in some signal transduction processes, impacting the signaling pathway and indirectly affecting the expression of downstream gene SAGs.

The interaction between ethylene signaling and various plant hormones or nutrient-specific repressive signals plays an important role in coordinating plant growth and survival in complex and dynamic environments. Ethylene, known as a senescence-promoting hormone, has been shown in a previous study to cooperatively promote leaf senescence with salicylic acid [11,57]. Conversely, the function of ethylene/jasmonate and salicylic acid is antagonistic in plant immune responses [58] and apical hook formation [59]. Furthermore, in the regulation of responses to phosphorus (P) deficiency, the crosstalk between ethylene and low Pi-induced signals may control Pi homeostasis by affecting *PSI* gene expression, thereby promoting root hair development [60,61]. However, the relationship between ethylene signaling and phytic acid has not been previously reported. This study revealed that the expression of *AtMINPP*, regulated by EIN3 during leaf senescence, mediates the crosstalk between ethylene signaling transduction and phytic acid metabolism.

In the model species Arabidopsis, phytic acid is primarily stored in the embryo [62]. Upon germination, phytic acid is degraded by phytase enzymes to release the phosphorus stored as phytate salts, supporting seedling growth [22]. While the leaf serves as a sink for nitrogen and mineral nutrients during the early stages of leaf development, it transitions into a nutrient source once leaf senescence begins. Nutrients such as potassium, nitrogen, phosphorus, sulfur, and metals are mobilized from yellowing leaves to support new growth or developing seeds [63,64,65,66]. We also propose a new hypothesis suggesting that the plant receives an ethylene signal, which in turn promotes the transcription factor EIN3 to regulate the expression of *AtMINPP.* This regulation facilitates phytic acid decomposition and transfers phosphate and mineral nutrients from old leaves to new leaves, thereby enabling plants to achieve internal substance circulation and reuse during leaf senescence in *Arabidopsis thaliana*. An intriguing and challenging aspect for future studies will be to determine whether phosphates from phytic acid metabolism play a key role in remobilization during leaf aging. This exploration could provide a novel insight into the sustainable utilization of phosphate rock resources and the development of environmentally friendly crops. 

In summary, our data support a proposed working model for the regulation of leaf senescence by AtMINPP (Figure 8). Ethylene receptors and CTR1 recognize ethylene, inhibiting CTR1 function and resulting in dephosphorylation of EIN2. Proteases then break down EIN2 into two parts. The EIN2–CEND protein translocates to the nucleus, activating key transcription factors EIN3/EIN3LIKE1 (EIL1)–dependent transcription. On one hand, EIN3/EIL1 can promote the expression of SAGs by directly activating the expression of *ORE1/NAC2*, influencing leaf senescence [67]. On the other hand, EIN3/EIL1 can enhance the expression of AtMINPP, which hydrolyzes phytic acid, thereby regulating phytic acid homeostasis in plants and modulating the expression of SAGs in leaf senescence.

## 4. Materials and Methods

### 4.1. Plant Materials and Growth Conditions

All mutants used in this study were derived from the Columbia background, with Col–0 serving as the control. *AtMINPP–OE* and *atminpp/+* were generated in our laboratory, and the homozygous lines were identified through PCR genotyping (primer pairs listed in Appendix A).

For seedlings grown on an agar plate, seeds were surface-sterilized using 75% ethanol (*v*/*v*) for 15 min, followed by three to five washes with sterile water. The sterilized seeds were then placed on a 1/2 MS agar plate and incubated at 4 °C for 72 h. Subsequently, they were transferred to a greenhouse maintained at 20–22 °C with 65% relative humidity under long-day conditions (16 h light/8 h dark) for 7 days. Seedlings with consistent growth were selected and transplanted into the soil. The light intensity during the growth period was maintained at 110 µmol photons m^−2^ sec^−1^ white light.

### 4.2. Sequence Analysis and Alignment

The conserved domains of AtMINPP and other proteins were deduced using NCBI conserved domain searches (http://www.ncbi.nlm.nih.gov/Structure/cdd/wrpsb.cgi, accessed on 26 February 2024) and HomoloGene (https://www.ncbi.nlm.nih.gov/homologene/37980, accessed on 26 February 2024). The polygenetic tree was constructed using the Neighbor-Joining method with 1000 bootstrap replicates using MEGA11 software (Version 11.0.13) [68]. The figures next to the branch demonstrate the result of 1000 bootstrap repeats expressed in percentage. Multiple alignments were created using CLUSTALW (https://www.ebi.ac.uk/jdispatcher/, accessed on 27 February 2024) [69], with minor manual adjustments as necessary to optimize the alignment.

### 4.3. Acid Phosphatase Activity, Phytase Activity, and Phytic Acid Content Assay

The full CDS of AtMINPP were cloned into the pET28a–MBP vector. The recombinant plasmid was transferred to *Escherichia coli* BL21 (DE3) and induced by isopropyl β-D-1thiogalactopyranoside (IPTG) for recombinant protein expression [70]. The acid phosphatase activity was assayed by a commercially available kit (Beyotime, Shanghai, China) according to the manufacturer’s instructions. In brief, the incubation mixture consisted of 40 μL of *p*-nitrophenyl phosphate solution, pH 4.8, and 40 μL recombinant purified proteins solution. The mixtures were incubated for 10 min at 37 °C, the enzymic hydrolysis was stopped by the addition of 160 μL of NaOH solution, and the absorbance value was determined at 405 nm. One unit of enzyme activity is the amount of enzyme which hydrolyses 1 μM of *p*-nitrophenyl phosphate in 1 min at 37 °C. The phytase activity was assayed by a phytase assay kit (Michy Biology, Suzhou, China). In brief, 30 μL of recombinant purified protein solution was added to a centrifuge tube containing 120 μL of sodium phytate solution and mixed well. The mixtures were incubated for 30 min at 37 °C, then 95 °C for 10 min, and then stopped to cool to room temperature. Immediately, 150 μL ammonium molybdate chromogenic agent was added and let stand at 37 °C for 15 min; then centrifuged for 5 min, 10,000 g, room temperature. Then, 200 μL of supernatant was used to determine the absorbance value at 700 nm. One unit of phytase activity is 1 μM of inorganic phosphorus and is released per minute from 5 mmol/L sodium phytate solution per milliliter of liquid. Determination of phytic acid content was assayed by a phytic acid content assay kit (Sinobestbio, Shanghai, China). In brief, 0.05 g of the sample was weighed, 1 mL of Na_2_SO_4_–HCl solution was added, and the sample was shaken for 2 h. Then, 8000 g was centrifuged at 25 °C for 10 min; 0.5 mL of supernatant was added to 0.5 mL of NaOH solution, mixed well, and let stand at 4 °C for 2 h. This was centrifuged, then 100 μL of supernatant was added to 900 μL of NaCl solution, mixed well, and 750 μL was added to 250 μL of ferric chloride-sulfosalicylic acid reaction solution. The absorbance was measured at 500 nm after full mixing.

### 4.4. Measurement of Chlorophyll Content

The measurement of chlorophyll content followed the method described by Sartory and Grobbelaar [71]. In brief, the fifth to eighth leaves were detached and extracted with 95% (*v*/*v*) ethanol. The absorbance was then measured at 663 nm and 649 nm. The chlorophyll content was calculated using the formula: (18.08 × A649 + 6.63 × A665) × dilution factor × g^−1^ fresh weight. Each experiment included a minimum of three biological replicates.

### 4.5. Chlorophyll Fluorescence Measurement of Fv/Fm

Chlorophyll fluorescence measurements were conducted using a WALZ IMAGING PAM system (Heinz Walz GmbH, Nuremberg, Germany). Prior to the measurements, the plant materials were dark-adapted for 20 min. The maximum (Fm) and minimum (F_0_) fluorescence values, as well as the maximal photochemical efficiency of PSII (Fv/Fm), were then recorded. Each experiment was performed with a minimum of three biological replicates.

### 4.6. Measurement of Ion Leakage

Ion leakage was assessed following a published method with minor adjustments [56]. For each experiment, a minimum of 12 leaves were immersed in 20 mL of deionized water for 30 min, and the conductivity was subsequently measured using a conductivity meter. To determine the total conductivity, the samples were boiled for 15 min. In the case of treatments (water, ACC, AgNO_3_, and AVG), the leaves were first subjected to the respective treatment, followed by a wash with deionized water prior to measurement. Each experiment included a minimum of three biological replicates.

### 4.7. Real-Time Quantitative PCR Analysis

Total RNA was extracted using MagZol Reagent (Magen, Guangzhou, China) following the manufacturer’s instructions. First-strand cDNA synthesis was performed using the Mon-Script^TM^ RTIII AII-in-One Mix with dsDNase (Monad, Suzhou, China), and the synthesized cDNA was subsequently employed as a template for quantitative RT-PCR (qRT-PCR) using ChamQ Universal SYBR qPCR Master Mix (Vazyme, Nanjing, China). Gene-specific primers utilized for qRT-PCR are listed in Appendix A. The internal control *TUB2* was an internal control for normalization. Each RT-qPCR was carried out with at least three biological replicates, each of which represented the mean of four technical replicates.

### 4.8. Yeast One-Hybrid Library Screening, Yeast One-Hybrid, and ChIP Assay

Total RNA was extracted from Arabidopsis leaves following the instructions provided in the user manual of the Matchmaker Gold Yeast One-Hybrid Library Screening System Kit (Clontech, Beijing, China). The AtMINPP promoter was constructed in a pAbAi vector, as described by Yang et al. [72]. For library screening, the Yeastmaker Yeast Transformation System 2 User Manual (Clontech) protocol was followed, and the screening was performed on synthetic dextrose (SD)/−Leu + AbA^100^ plates in a 30 °C incubator for 4 days. Single colonies were selected and amplified by PCR, and their DNA sequences were determined. The colonies encoding TFs were considered potential candidates for further investigation. 

To verify the interactions indicated by the library screening and identify individual interactions between a single TF and the target gene promoter, the Yeast one-hybrid assay was employed. This assay utilized the Matchmaker Gold Yeast One-Hybrid Library Screening System (Clontech, Beijing, China). The full-length sequences of EIN3 candidates were subcloned into the pGADT7 AD vector (primers can be found in Appendix A), and the interaction analyses were conducted following the manufacturer’s protocol.

Chromatin immunoprecipitation (ChIP) was performed as previously described [73]. Ten-day-old seedlings were fully ground in liquid nitrogen and cross-linked in 1% formaldehyde (Sigma-Aldrich, Beijing, China) for 10 min on ice. The chromatin was then pelleted by centrifugation and sonicated to obtain DNA fragments of approximately 500 bp. The lysate was pre-cleared with 50 mL protein-A agarose beads/salmon sperm DNA (Millipore, Shanghai, China) for 1 h and then incubated overnight with anti-MYC antibodies (Abcam, Shanghai, China). The anti-MYC antibody was used at a concentration of 5 mg per 0.5 g seedling tissue. The bound chromatin was purified using columns from the Qiagen plasmid extraction kit. Real-time PCR was performed on the input, no antibody control, and antibody-bound DNA in triplicates. Three biological replicates were conducted to ensure reproducibility. The specific primers used in ChIP-PCR are listed in Appendix A, and *TUB2* was used as a reference gene.

### 4.9. Treatment with ACC, AgNO_3_, and AVG

The experiment was conducted with slight modifications based on a previous study [56]. Col–0 seedlings were grown on 1/2 MS with 0.6% agar Petri dishes for 2 weeks and used for the ACC treatment. A 50 mL solution containing 1/2 MS liquid culture with 20 µM ACC was poured into the plates, while the solution without ACC served as the control. The materials were incubated under long-day conditions and harvested at specified time points for RNA isolation.

To examine the leaf senescent phenotype, the fifth to eighth rosette leaves of 4-week-old Col–0, *AtMINPP–OE*, and *atminpp/+* plants were detached for ACC, AgNO_3_, and AVG treatments. The detached leaves were placed into Petri dishes with a 6-well plate, and each well was supplemented with a 5 mL solution of ACC (100 µM), AgNO_3_ (50 µM), or AVG (50 µM). Leaves treated with water served as controls. The plates were sealed with parafilm and wrapped with a double layer of aluminum foil before being placed in a greenhouse set at 20–22 °C. For treatment, the leaves were treated for 1 h, washed three times with water, transferred to a new 6-well plate filled with water, and rewrapped. All the leaves were photographed and used for chlorophyll extraction at specific time intervals.

## Figures and Tables

**Figure 1 ijms-25-08969-f001:**
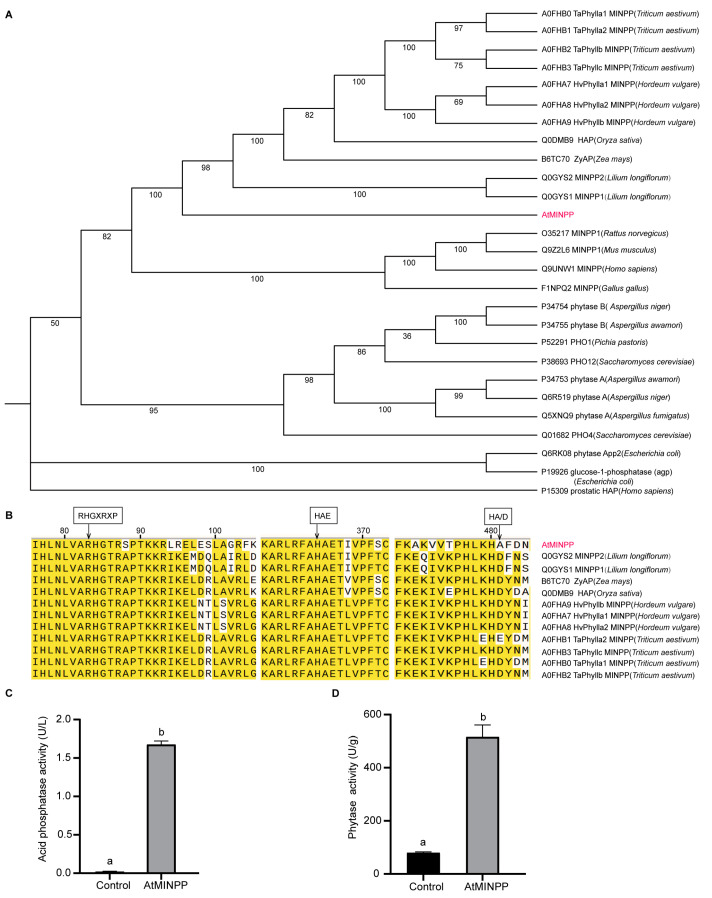
Identification of *AtMINPP* in *Arabidopsis thaliana*. (**A**) Phylogenetic relationship between *Arabidopsis thaliana* multiple inositol phosphate phosphatases (MINPPs) and other known histidine acid phosphatase (HAP), acid phosphatase (AP), and phosphatases from plants, microbes, animals, and humans. (**B**) Multiple alignment of amino acid sequences between AtMINPP and other multiple inositol phosphate phosphatases, HAP phytases, and acid phosphatases around the RHGXRXP, HAE, and HD motifs of the active site. Yellow highlight indicates the same amino acid sequence among the twelve species, red font indicates the amino acid sequence of AtMINPP. (**C**,**D**) Acid phosphatase activity (**C**) and phytase activity assay (**D**) of the recombinant protein. MBP protein as control. Error bars represent the means ± SD (n = 3). The different lowercase letters above the columns indicate significant differences according to Student’s *t*-test (*p* < 0.01).

**Figure 2 ijms-25-08969-f002:**
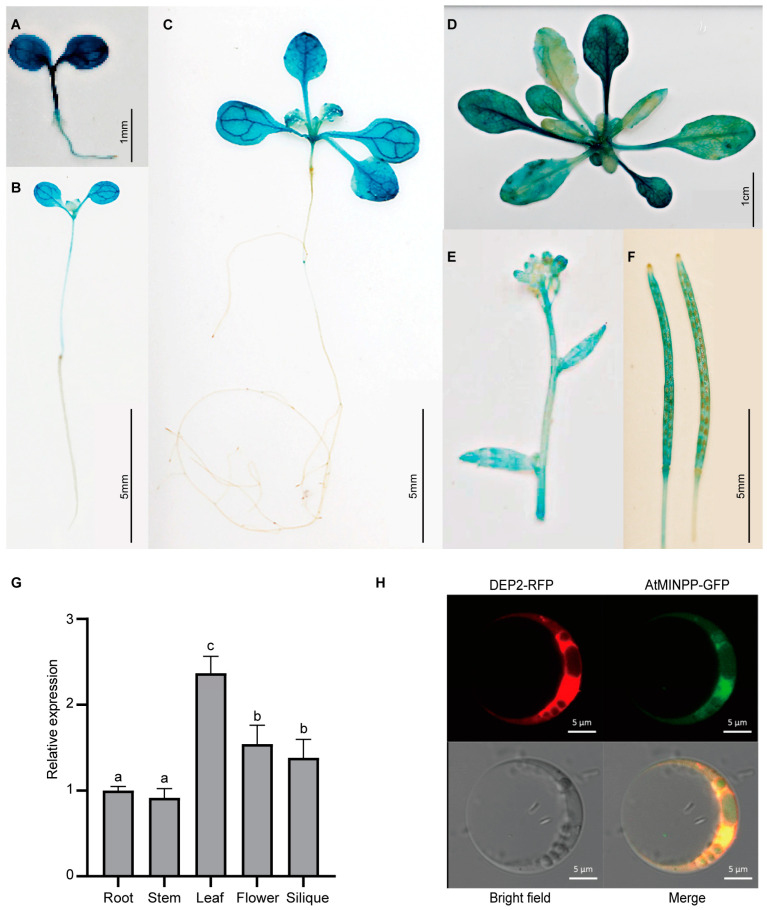
Expression patterns of the *AtMINPP* gene. (**A**–**F**) Tissue-specific expression patterns of *AtMINPP::GUS.* (**A**) A 3-day-old seedling; (**B**) 7-day-old seedling; (**C**) 14-day-old seedling; (**D**) 22-day-old mature plant; (**E**) flower buds, stems, and cauline leaves; (**F**) siliques. (**G**) Expression levels of the *AtMINPP* gene in different tissues were detected by RT-qPCR. Error bars represent the means ± SD (n = 3). The different lowercase letters above the columns indicate significant differences according to one-way ANOVA (*p* < 0.05). *TUB2* was used as an internal reference. (**H**) The subcellular localization of AtMINPP in Arabidopsis protoplasts.

**Figure 3 ijms-25-08969-f003:**
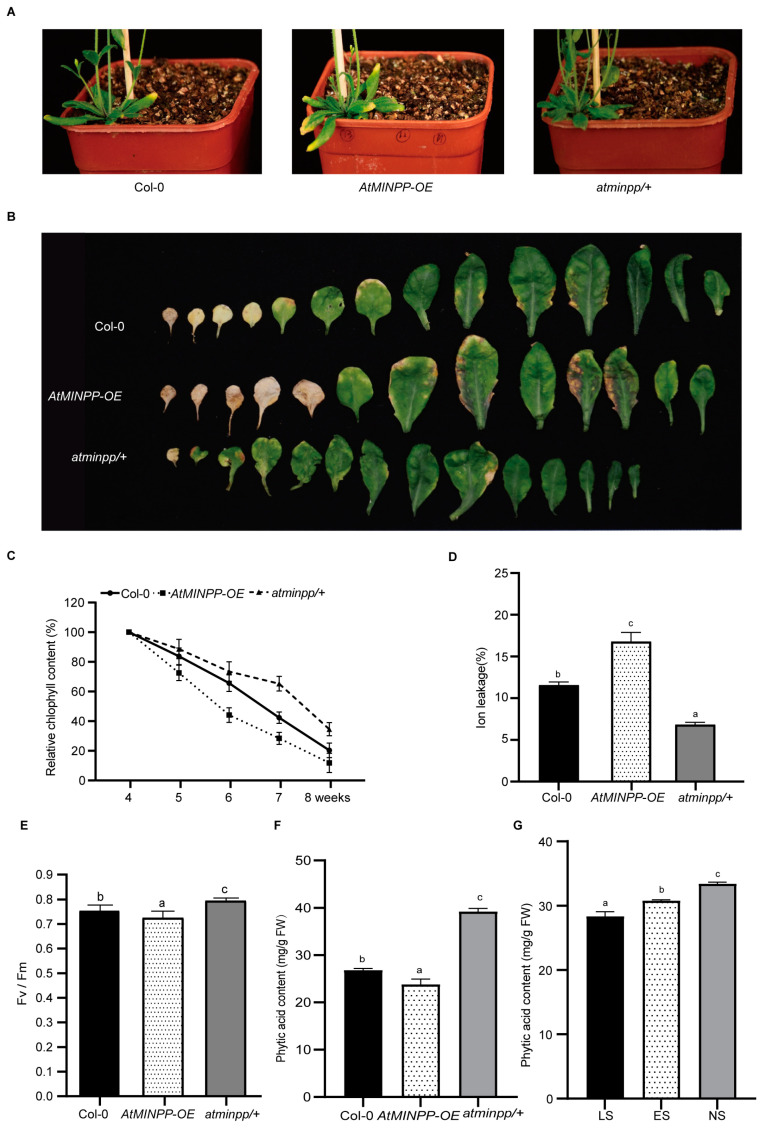
Phytic acid metabolism regulated by AtMINPP is involved in leaf senescence. (**A**) The leaf senescence phenotypes of 5-week-old plants. (**B**) The leaf senescence phenotypes of detached rosette leaves from 6-week-old plants. (**C**) Relative chlorophyll content in fifth to eighth leaves of the three genotypes at the indicated ages; those in 4-week-old plants were designated as 100%. (**D**,**E**) Analysis of ion leakage (**D**) and Fv/Fm values (**E**) from detached fifth to eighth leaves of 5-week-old plants. (**F**,**G**) Analysis of the phytic content from detached fifth to eighth leaves of 5-week-old plants (**F**) and those leaves at three different developmental stages (**G**). NS, nonsenescent leaves; ES, early senescent leaves; LS, late senescent leaves. Error bars in C-G represent the means ± SD (n = 3). Different lowercase letters above the columns indicate significant differences according to one-way ANOVA (**C**–**E**): *p* < 0.05; (**F**,**G**), *p* < 0.01).

**Figure 4 ijms-25-08969-f004:**
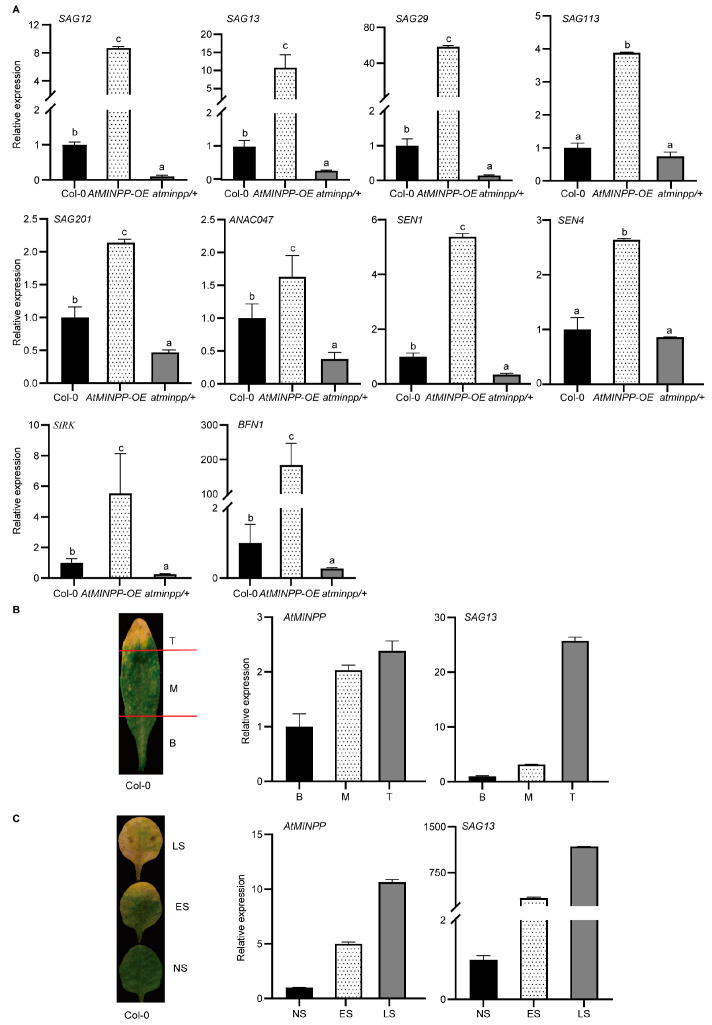
AtMINPP plays a key role in the regulation of SAGs. (**A**) Expression levels of SAGs in the fifth to eighth leaves of 5-week-old Col–0, *AtMINPP–OE*, and *atminpp/+* plants detected by RT-qPCR. (**B**,**C**) Expression levels of *AtMINPP* in different parts (**B**) and different developmental stages (**C**) of the sixth leaves of Col–0. *SAG13* was used as a positive control. B, base part; M, middle part; T, tip part. NS, nonsenescent leaves; ES, early senescent leaves; LS, late senescent leaves. Error bars represent the means ± SD (n = 3). Different lowercase letters above the columns indicate significant differences according to one-way ANOVA (*p* < 0.05). *TUB2* was used as an internal reference in (**A**–**C**).

**Figure 5 ijms-25-08969-f005:**
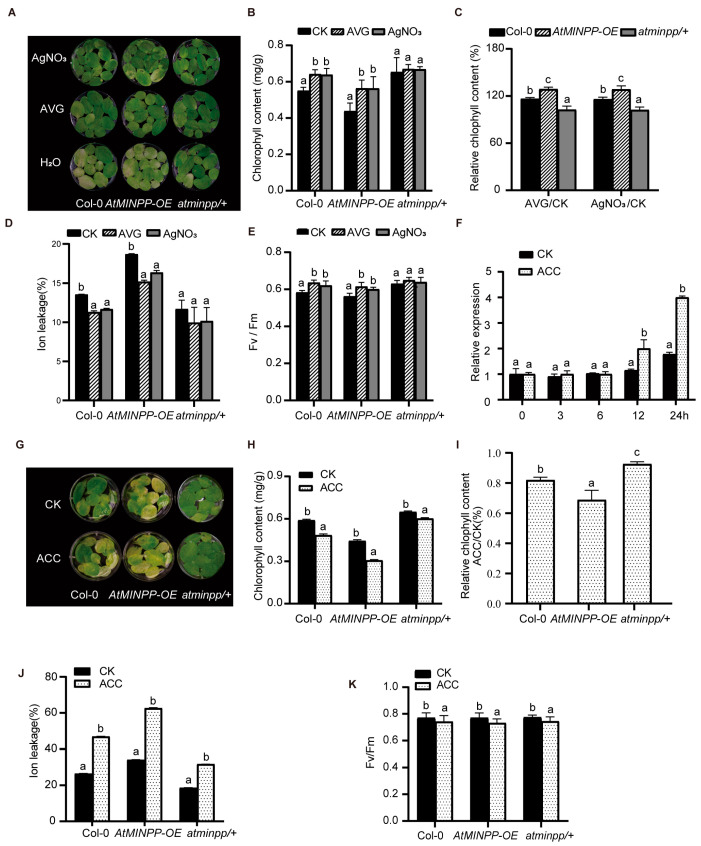
AtMINPP participates in ethylene-induced leaf senescence. (**A**,**G**) The senescent phenotypes of detached leaves (fifth to eighth) of Col–0, *AtMINPP–OE*, and *atminpp/+* treated with water (CK), 50 μM AgNO_3_, 50 μM AVG, and 100 μM ACC under dark conditions. (**B**,**H**) Chlorophyll content of the three genotype leaves with indicated treatment (FW, fresh weight). (**C**,**I**) Chlorophyll content ratio of the three genotype leaves with indicated treatment. AgNO_3_/CK, AVG/CK, and ACC/CK indicate the relative ratio between leaves treated with AgNO_3_, AVG, or ACC and the controls for each genotype, respectively. (**D**,**J**) Ion leakage of the three genotype leaves with indicated treatment. (**E**,**K**) Fv/Fm values of the three genotype leaves with indicated treatment. (**F**) Expression level of *AtMINPP* in the 2-week-old Col–0 seedlings treated with ACC for indicated time. Error bars represent the means ± SD (n = 3). Different lowercase letters above the columns indicate significant differences according to two-way ANOVA (*p* < 0.05).

**Figure 6 ijms-25-08969-f006:**
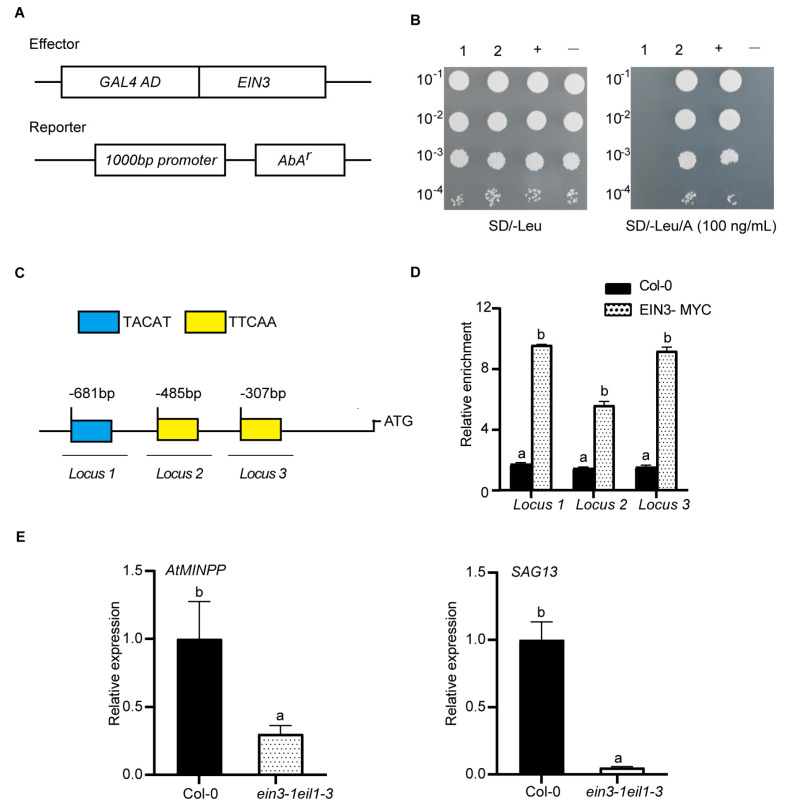
EIN3 is associated with the *AtMINPP* promoter and transcriptionally activates its expression. (**A**) Effector and reporter constructs used in the yeast one-hybrid assay. (**B**) Yeast one-hybrid assay detected the interaction of the *AtMINPP* promoter with EIN3. (**C**) The promoter structure of *AtMINPP.* Primers used for ChIP-PCR were specific to the promoter regions containing EIN3-binding sites. *Locus1* to *Locus3* with short lines indicate fragments for ChIP analysis. (**D**) The ChIP-quantitative PCR analysis for EIN3. Col–0 and EIN3–MYC represent enrichment abundance from Col–0 and *35S::EIN3–MYC*. *Locus1* to *Locus3* indicated the detected individual fragments of *AtMINPP* promoters. A sequence with no predicted binding sites of *AtMINPP* was used as negative control. (**E**) Transcript levels of *AtMINPP* between Col–0 and *ein3–1eil1–3* by RT-qPCR. *SAG13* was used as a positive control. Error bars represent the means ± SD (n = 3). Different lowercase letters above the columns indicate the significant differences according to two-way ANOVA ((**D**): *p* < 0.01) and Student’s *t*-test ((**E**): *p* < 0.05), respectively.

**Figure 7 ijms-25-08969-f007:**
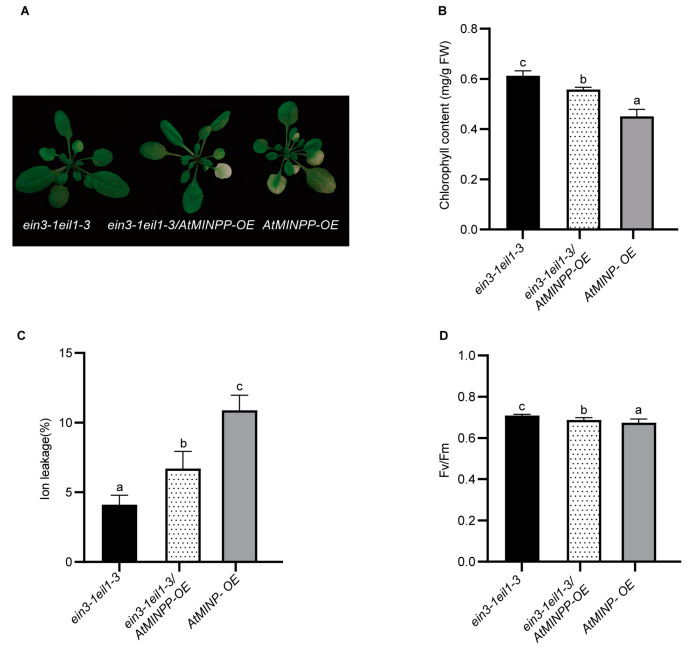
EIN3 promotes the expression of *AtMINPP*. (**A**) The senescent phenotypes of rosette leaves from 5-week-old plants. (**B**) Chlorophyll contents in the leaves of 5-week-old plants (FW, fresh weight). (**C**) Ion leakage rates of the detached leaves of 5-week-old plants. (**D**) Fv/Fm values in the detached leaves of 5-week-old plants. In (**B**–**D**), the fifth to eighth leaves from Col–0, *AtMINPP–OE*, and *atminpp/+* were used for each experiment. Error bars in (**B**–**D**) represent the means ± SD (n = 3). Different lowercase letters above the columns indicate the significant differences according to one-way ANOVA ((**B**,**C**): *p* < 0.05; (**D**): *p* < 0.01).

**Figure 8 ijms-25-08969-f008:**
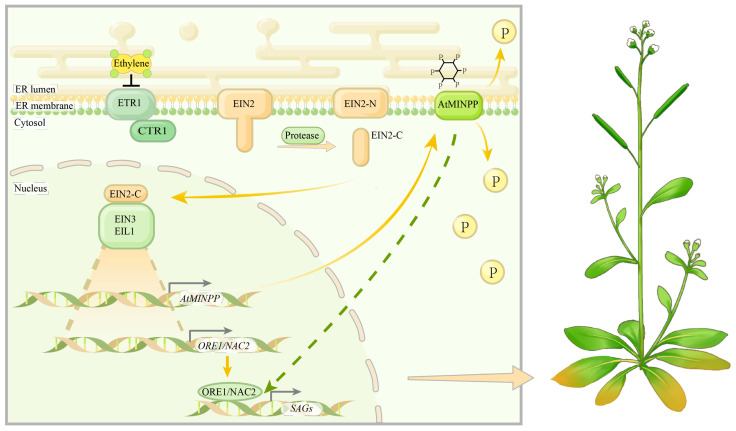
A working model of the functioning of AtMINPP in ethylene-mediated leaf senescence. Yellow arrowheads represent positive regulation; black blunt arrows represent negative regulation; green dashed lines represent indirect regulation.

## Data Availability

Data are contained within the article.

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
