# Peer review of "The AtMINPP Gene, Encoding a Multiple Inositol Polyphosphate Phosphatase, Coordinates a Novel Crosstalk between Phytic Acid Metabolism and Ethylene Signal Transduction in Leaf Senescence"

_ijms, 2024, doi:10.3390/ijms25168969_

Round 1
Reviewer 1 Report
Comments and Suggestions for Authors
In this manuscript, the authors provided data elucidating the role and possible mechanism of AtMINPP in leaf senescence. The results are helpful for understanding the regulation of leaf senescence and the ethylene signaling cascade. The story is novel and interesting. Nevertheless, necessary revisions should be performed.
1. Some figures should be re-organized. (1) Figure 1, the asterisks above the column are too small to be recognized. In the figure legend, line119, “comtrol” should be corrected to “control”. (2) Figure 2 should be reformatted to make the layout more compact. In the figure legend, line 144, “Expression levels of the AtMINPP gene in different tissues by RT-qPCR” should be revised to “Expression levels of the AtMINPP gene in different tissues were detected by RT-qPCR.” (3) Figure 3 should be reformatted to make the layout more compact. In the figure legend, the sentences “Error bars……” and “Different lowercases………” were repeated several times. It is unnecessary. (4) Figure 4 should be reformatted to make the layout more compact. In figure A, some data points are invisible, but lowercase letters were marked above them. The figure should be re-plotted to make these columns visible. In the figure legend, the sentence “Error bars……” was repeated twice. (5) Figure 5 should be reformatted to make the layout more compact. In the figure legend, line 252, “for 0h, 3h, 6h, 12h and 24h. CK, control plants; ACC, ACC treated plant” should be deleted since the parameters had been marked in figure F. (6) Figure 6, in the figure legend, the sentence “Error bars……” was repeated twice. (7) Figure 7, in the figure legend, it was noted that “Different lowercase letters above the columns………”; however, I can only find asterisks in the figures.
2. In some figures, data was analyzed using ANOVA, but data points were marked using asterisks. It is recommended to use lowercase letters instead of asterisks for better data presentation and understanding.
3. Line 270, “Figure 7E” should be corrected to “Figure 6E”.
4. In Lines 187-188, the sentence “we examined the expression profile of AtMINPP” is not understandable. In fact, it is gene expression profile in different genetic backgrounds of AtMINPP. The sentence should be revised.
5. The authors should discuss how phytic acid levels affect the expression of SAGs. Although they provided strong evidence about how EIN3 affect the expression of AtMINPP, there is still no evidence about the effect of phytic acid decomposition products on the expression of any genes.
Reviewer 2 Report
Comments and Suggestions for Authors
The Authors have shown that multiple inositol phosphate phosphatase MINPP gene (At1G09870) plays a role il leaf senescence in Arabidopsis thaliana via ethylene signal transduction pathway – the gene is regulated by ethylene responsive transcription factor EIN3; they observed that overexpressor line shows early senescence, upregulation of SAG’s and loss of chlorophyll, while heterozygous loss of function mutant line shows an opposite phenotype. The work describes a function of the next model plant A. thaliana gene, what makes it a valuable contribution to our basic knowledge of plant biology. It would be of interest to check to what extent modulation of activity of this gene can coordinate phosphate uptake, flowering and senescence.
A few minor remarks:
Line 82: “phytases belong” – it seems it should be belonging or which belong.
Fig1C and D and description in text – it is not clear for me are there units/ liter of fresh leaf? tissue? extract? And per gram of purified enzyme?
Line 124 – iw would suggest to use a name pod instead of legume since these organs are clearly different, so their names are not synonymous.
Comments on the Quality of English LanguageLine 82: “phytases belong” – it seems it should be belonging or which belong.
Fig1C and D and description in text – it is not clear for me are there units/ liter of fresh leaf? tissue? extract? And per gram of purified enzyme?
Line 124 – iw would suggest to use a name pod instead of legume, since these organs are clearly different, so their names are not synonymous.
